

# Conformal boundary conditions for a 4d scalar field

### Lorenzo Di Pietro[1,2⋆], Edoardo Lauria[3†] and Pierluigi Niro[4‡]

**1** Dipartimento di Fisica, Università di Trieste, Strada Costiera 11, I-34151 Trieste, Italy
**2** INFN, Sezione di Trieste, Via Valerio 2, I-34127 Trieste, Italy
**3** LPENS, Département de physique, École Normale Supérieure - PSL,
Centre Automatique et Systèmes (CAS), Mines Paris - PSL,
Université PSL, Sorbonne Université, CNRS, Inria, 75005 Paris
**4** Mani L. Bhaumik Institute for Theoretical Physics, Department of Physics and Astronomy,
University of California, Los Angeles, CA 90095, USA

⋆ ldipietro@units.it , † edoardo.lauria@minesparis.psl.eu , ‡ pniro@physics.ucla.edu

## Abstract

We construct unitary, stable, and interacting conformal boundary conditions for a free massless scalar in four dimensions by coupling it to edge modes living on a boundary. The boundary theories we consider are bosonic and fermionic QED$_3$ with $N_f$ flavors and a Chern-Simons term at level $k$, in the large-$N_f$ limit with fixed $k/N_f$. We find that interacting boundary conditions only exist when $k \neq 0$. To obtain this result we compute the $\beta$ functions of the classically marginal couplings at the first non-vanishing order in the large-$N_f$ expansion, and to all orders in $k/N_f$ and in the couplings. To check vacuum stability we also compute the large-$N_f$ effective potential. We compare our results with the the known conformal bootstrap bounds.

# 1 Introduction

What are the possible conformal boundary conditions, or BCFTs, for a given bulk conformal field theory (CFT)? It is not known how to constructively answer this question when the bulk dimension $d$ is larger than two.[1] Besides the interest from the point of view of the formal aspects of quantum field theory, it is also a question of practical relevance, e.g. to give predictions for the possible "surface transitions" of a given second-order phase transition when it is realized in an enclosed region of space. On first thought, this problem might seem as hard as the classification of one-lower dimensional CFTs. Indeed, starting with any given conformal boundary condition and any conformal "edge modes" localized on the boundary, the latter can be coupled to the bulk via some relevant deformation. By then allowing the boundary Renormalization Group (RG) flow to settle in an IR fixed point, this construction seemingly produces a plethora of new BCFTs. However this formal argument does not rule out the possibility that many of these RG flows actually settle in the same IR fixed point, or that at such IR fixed point the bulk-boundary coupling vanishes, making the set of boundary conditions much more restricted.

A possible approach to shed some light on the original question is to restrict to a bulk CFT that is as simple as possible. Even for a single free massless scalar, the problem reveals many curious surprises, suggesting that we are just scratching the surface of the matter. In this case, the argument above can be phrased by starting with free boundary conditions, i.e. Neumann or Dirichlet, and by coupling the boundary mode of the scalar to local CFTs on the boundary. Here "free" or "interacting" boundary condition refers to whether correlation functions of local boundary operators are products of two point functions. The non-trivial problem is then to find examples in which the IR fixed point is *not* again of the form of a free boundary condition with a decoupled local sector. The purpose of this note is precisely to exhibit examples of such interacting conformal boundary conditions in the case of a four-dimensional bulk.[2]

A numerical bootstrap study of BCFTs for a single free massless scalar in three and four dimensions was carried out in references [6, 9], which have shown that the space of possible such theories is highly constrained.[3] In particular, figure 2 of [9] shows a prominent kink close to the Neumann boundary condition and a very small spin-two gap elsewhere. While the nature of that kink remains somewhat mysterious, the very small spin-two gap suggests

---

[1]In 2d a classification is known for boundary conditions of rational CFTs that preserve the chiral algebra (see [1–4] for reviews), and a partial classification exits for the free compact boson or fermion [5].

[2]For a 3d bulk, an example was found in [6]. Other interesting examples of perturbative constructions of boundary conditions for a free scalar field can be found in [7,8].

[3]Systematic studies of the space of higher co-dimension conformal defects for free theories can be found in [10] (for the free massless scalar) and in [11] for the 4d Maxwell field.

in particular that, if they exist, unitary, local (in the sense of [9]) and interacting conformal boundary conditions near the Dirichlet end should be almost decoupled from the bulk. It is then reasonable to expect that points in this portion of the allowed region should be within reach of perturbative techniques.

Motivated by this observation, in this note we look for perturbative BCFTs fixed points for a free massless scalar in four dimensions. As boundary degrees of freedom we choose three-dimensional quantum electrodynamics (QEDs) with many flavors – either bosonic or fermionic. In addition to our goal, this type of boundary conditions can also be applied to the long-distance physics of the edge states of a three-dimensional symmetry-protected topological phase at the four-dimensional order-disorder quantum critical point (see e.g. [12] and references therein). We write down all possible classically marginal interactions between the bulk and the boundary, assuming that all the relevant ones are tuned to zero. The possible marginal couplings depend on whether we fix Neumann or Dirichlet condition for the bulk field, and whether or not the matter degrees of freedom on the boundary have quartic self-interactions. Moreover as it will become clear, in order to find interacting conformal boundary conditions we need to include a Chern-Simons (CS) term $k$ for the boundary $U(1)$ gauge field.

Given that we are localizing these gauge theories on a boundary of spacetime, we can equivalently realize them as decoupling limits of the boundary matter fields interacting with a bulk Maxwell gauge field [13–16]. This invites to consider a slightly more general setup, in which the matter fields localized on the 3d boundary interact with both a bulk scalar field $\Phi$ and a bulk Maxwell field $A_\mu$, i.e.

$$S_{\text{bulk}} = \int_{y \geq 0} d^3x\, dy \left[ \frac{1}{2}(\partial_\mu \Phi)^2 + \frac{N_f}{4\lambda}\left( F_{\mu\nu}F^{\mu\nu} + i\frac{\gamma}{2}\epsilon_{\mu\nu\rho\sigma}F^{\mu\nu}F^{\rho\sigma} \right) \right]. \tag{1}$$

Here $N_f/2$ is the number of complex boundary scalars or Dirac fermions, whose actions (along with the boundary conditions for $\Phi$ and $A_\mu$) will be specified below. In the formula above the (flat) boundary sits at $x^\mu = (y = 0, x^a)$ and we have chosen an orientation of the half-space such that $\epsilon_{abcy} = \epsilon_{abc}$, with latin indices labeling parallel directions with respect to the boundary. Unlike the boundary interactions, the bulk parameters $\lambda$ and $\gamma$ are exactly marginal. The parameter $\gamma$ is related to the bulk $\theta$-angle as $\lambda\theta = 4\pi^2 N_f \gamma$.

We compute the $\beta$ functions for the boundary couplings, at the leading order in the $1/N_f$ expansion but exactly in the couplings, and with arbitrary $\gamma$ and $\lambda$. We find RG fixed points that correspond to unitary BCFTs. At the end of the calculation we decouple the bulk Maxwell field by taking the limit of large $\lambda$, in such a way that by electric-magnetic duality the setup is equivalent to QED$_3$ coupled to the bulk scalar. The value of $\gamma/\lambda$ in this limit determines the ratio $k/N_f$. In particular we find that only when $k/N_f$ lies in certain ranges the interacting BCFTs for the scalar field are unitary, that is they correspond to real values of the fixed points for the marginal couplings.

After finding non-trivial and real fixed points we address the question of the stability of their vacuum. We do this by computing the effective potentials as functions of the boundary couplings at leading order at large $N_f$, both for bosonic and fermionic matter fields. The stability condition constrains the allowed values of the couplings, and comparing with their values at the fixed points we find further restrictions on the possible values of the free parameter $k/N_f$. The results of our analysis are summarized in figures 4 (for bosons) and 9 (for fermions). One immediate conclusion is that the interacting boundary conditions only exist for $k \neq 0$: restricting to $k = 0$ one would find that in the fixed points the bulk scalar always decouples from the boundary. This demonstrates that finding interacting BCFTs is not as simple as the naive argument above seemed to suggest.

Finally, we compare the results with the numerical bootstrap bounds. As a consistency check, we find that our perturbative results lie inside the allowed regions. We provide at least

three checks that this is the case, by computing the anomalous dimensions of the first two lowest-lying singlet-scalar operators in the boundary spectrum at order $1/N_f$, by computing the anomalous dimension of the leading spin-two operator on the boundary at order $1/N_f$, and by computing the two-point function of the displacement operator when $N_f = \infty$.

## 2 Bosons on the boundary

We consider a 4d bulk scalar field $\Phi$ and bulk Maxwell field $A_\mu$, with Dirichlet and Neumann boundary condition (respectively), coupled to $N_f/2$ boundary complex scalars $z^m$, with $m = 1, \ldots, N_f/2$. The action is given by

$$S = S_{\text{bulk}} + \int_{y=0} d^3x \left[ (\mathscr{D}_a z^m)^\dagger (\mathscr{D}_a z^m) + \frac{2g}{\sqrt{N_f}} \partial_y \Phi z^{\dagger m} z^m + \frac{8h}{N_f^2} (z^{\dagger m} z^m)^3 \right]. \qquad (2)$$

In the formula above $\mathscr{D}_a \equiv \partial_a + iA_a$ is the covariant derivative, and $S_{\text{bulk}}$ is given in (1). For generic values of the couplings, the continuous part of the boundary global symmetry is $SU(N_f/2) \times U(1)^2$, where the first factor is a flavor symmetry and the second one comes from the currents $\epsilon_{abc} F^{bc}$ and $F_{ya}$. Note that the free Neumann boundary condition sets $F_{ya} = 0$, but with boundary degrees of freedom one instead has a "modified Neumann" condition, that identifies $F_{ya}$ with the $U(1)$ current of the boundary matter, so that indeed there is an additional conserved current. For $\gamma = 0$ the theory further enjoys parity symmetry.

In the large-$N_f$ limit, with $g$, $h$, $\lambda$, $\gamma$ held fixed, the theory (2) interpolates between different bulk/boundary decoupling limits that correspond to different 3d local CFT sectors. The bulk gauge field decouples when the complex gauge coupling $\tau = \frac{2\pi i}{\lambda/N_f} + \frac{2\pi\gamma}{\lambda/N_f}$ is $i\infty$ or, by applying a bulk $SL(2, \mathbb{Z})$ transformation, whenever it attains rational values. At these rational values, the boundary $U(1)$ current to which the bulk gauge field couples is gauged by emergent 3d gauge fields [13, 16]. This is true even at finite $N_f$, since $\tau$ and $\bar{\tau}$ do not run. The bulk scalar field decouples when either $g = 0$ or $g = \infty$, and in the latter case the scalar sector in the action (2) admits a dual description as a large-$N_f$ critical vector model [17]. Combining these results we obtain the following decoupling limits (see fig. 1):

I. $N_f$ real scalars for $g = 0$ and $\lambda = 0$, with a sextic interaction $h$.

II. The critical $O(N_f)$ model for $g = \infty$ and $\lambda = 0$.

III. Critical bosonic QED$_3$ with CS level $k$ and $N_f/2$ flavors of complex scalars, for $g = \infty$ and $\lambda = \gamma = \infty$, with $k/N_f = 2\pi\gamma/\lambda$ fixed.

IV. Tricritical bosonic QED$_3$ with CS level $k$ and $N_f/2$ flavors of complex scalars, for $g = 0$ and $\lambda = \gamma = \infty$, with $k/N_f = 2\pi\gamma/\lambda$ fixed, and a sextic interaction $h$.

Here "tricritical" refers to the relevant quartic interaction between the complex scalars tuned to zero. While $\lambda$ and $\gamma$ are always exactly marginal couplings, $g$ and $h$ are exactly marginal when $N_f = \infty$, and the decoupling limits mentioned are just special points within a family of conformal boundary conditions. At order $1/N_f$, both $g$ and $h$ develop non-trivial, $\lambda$- and $\gamma$-dependent $\beta$ functions, and the decoupling limits will be generically connected by RG flows.

### 2.1 Stability of marginal couplings

In this section we derive the stability bounds on the marginal couplings in the theory (2). At the classical level we clearly need $h > 0$, while $g$ is unconstrained.[4] At the quantum level we

---

[4] The sign of $g$ can be reabsorbed by a sign flip of $\Phi$, which is a global symmetry of the bulk theory.

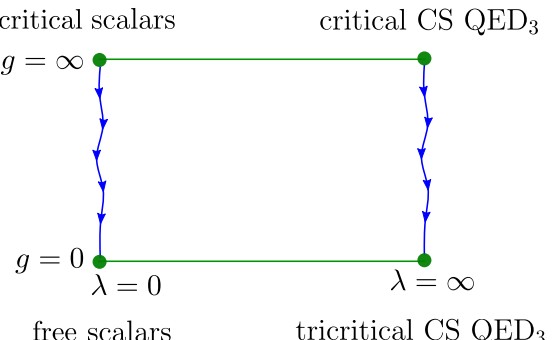

Figure 1: The four 3d scalar CFTs connected via the interactions to the bulk scalar (vertical lines) and the bulk gauge field (horizontal lines).

have to compute the large-$N_f$ effective potential. To this end we shall rewrite the boundary Lagrangian in (2) as

$$\mathscr{L} = \frac{1}{2}(\partial_a \varphi^i)(\partial_a \varphi^i) + \frac{g}{\sqrt{N_f}}\, \partial_y \Phi \, \varphi^i \varphi^i + \sigma\left(\frac{\varphi^i \varphi^i}{\sqrt{N_f}} - \rho\right) + \frac{h}{\sqrt{N_f}}\rho^3\,, \tag{3}$$

where now $\varphi^i$ are $N_f$ reals scalars, and $\sigma$ is a Lagrange multiplier which identifies the field $\rho$ with the composite operator $\varphi^i \varphi^i / \sqrt{N_f}$. Note that we can neglect the gauge field, as its contribution to the effective potential is subleading in the large-$N_f$ expansion.

Let us now turn on vacuum expectation values (vevs) for the fields that appear in eq. (3). As for the bulk scalar, as a consequence of the bulk equations of motion this is a harmonic function of the transverse coordinate, which should stay finite as $y \to \infty$. Hence:

$$\langle \Phi(x^a, y)\rangle = \sqrt{N_f}U\,, \qquad \langle \partial_y \Phi(x^a, y)\rangle = 0\,, \tag{4}$$

where $U$ scales as $\mathcal{O}(N_f^0)$. For the remaining fields we let

$$\varphi^i = \sqrt{N_f}v^i + \hat{\varphi}^i\,, \qquad \sigma = \sqrt{N_f}\Sigma + \delta\sigma\,, \qquad \rho = \sqrt{N_f}\eta + \delta\rho\,, \tag{5}$$

where $v^i$, $\Sigma$, and $\eta$ are vevs that scale as $\mathcal{O}(N_f^0)$. Upon plugging into (3) and performing the Gaussian integral over the $N_f$ fields $\hat{\varphi}^i$ we get for the effective potential

$$\mathscr{V}_{\text{eff}} = N_f\left(\Sigma(v^i v^i - \eta) + h\eta^3\right) + \frac{N_f}{2}\text{tr}\log\left(-\Box + 2\Sigma\right) + \mathcal{O}(N_f^0)\,. \tag{6}$$

The trace in the expression above can be performed in dimensional regularization to find

$$\mathscr{V}_{\text{eff}} = N_f\left(\Sigma(v^i v^i - \eta) + h\eta^3 - \frac{1}{12\pi}(2\Sigma)^{3/2}\right) + \mathcal{O}(N_f^0)\,, \tag{7}$$

which is real if $\Sigma \geq 0$. We can take derivative of $\mathscr{V}_{\text{eff}}$ with respect to $v^i$, $\Sigma$, and $\eta$ to find the gap equations

$$\Sigma v^i = 0\,, \qquad v^i v^i - \eta - \frac{(2\Sigma)^{1/2}}{4\pi} = 0\,, \qquad -\Sigma + 3h\eta^2 = 0\,. \tag{8}$$

Hence there are two classes of solutions, namely

$$\begin{aligned} v^i = 0\,, \quad &\Sigma \geq 0\,, \quad \eta = -\frac{(2\Sigma)^{1/2}}{4\pi} \leq 0\,, \quad \text{unHiggsed,}\\ v^i \neq 0\,, \quad &\Sigma = 0\,, \quad \eta = v^i v^i > 0\,, \quad\quad\quad \text{Higgsed.} \end{aligned} \tag{9}$$

We can further integrate out the auxiliary field $\sigma$, i.e. we plug the second of (8) back into the effective potential to get

$$\mathcal{V}_{\text{eff}} = N_f \left( \frac{8\pi^2}{3} (v^i v^i - \eta)^3 + h\eta^3 \right) + \mathcal{O}(N_f^0), \tag{10}$$

which shows that the unique solution to the gap equations $(v^i, \eta) = (0, 0)$ corresponds to a stable vacuum (i.e. to a global minimum of the effective potential) if

$$0 < h < \frac{8\pi^2}{3}. \tag{11}$$

The classical stability region is therefore restricted by quantum effects. Intuitively, this has to be the case since bosonic self-interactions are repulsive and tend to destabilize the vacuum.

Note that the bound (11) does not depend on $g$ and holds for any finite $g$. In the limit where $g = 0$ it matches the bound derived in [18], once we implement the relation $h_{there} = 2h_{here}$. In the limit $g \to \infty$ the bulk scalar field decouples and the resulting 3d theories are the ones where the quartic coupling flows to criticality. In such models the sextic coupling is not allowed, since the equations of motion for the Hubbard-Stratonovich field imply that $\varphi^i \varphi^i = 0$. As a consequence, there is no stability bound on $h$ and the theory is always stable at the leading order in the large-$N_f$ expansion, see [18].

## 2.2 RG analysis

In this section we present the $\beta$ functions for the marginal couplings, as well as the anomalous dimensions for a few operators of the theory in eq. (2). We work at leading non-trivial order in the $1/N_f$ expansion. We recall that in our large-$N_f$ limit the couplings $g, h, \lambda, \gamma$ are held fixed (hence the $\theta$-term is large and scales as $N_f$). Feynman rules are collected in appendix A.

### 2.2.1 Exact propagators at large $N_f$

The first quantities that we shall compute are the large-$N_f$ boundary propagators of $\partial_y \Phi$ and $A_a$. These are obtained by resumming the geometric series of the 1PI bubbles connected by tree-level propagators, as depicted in fig. 2.

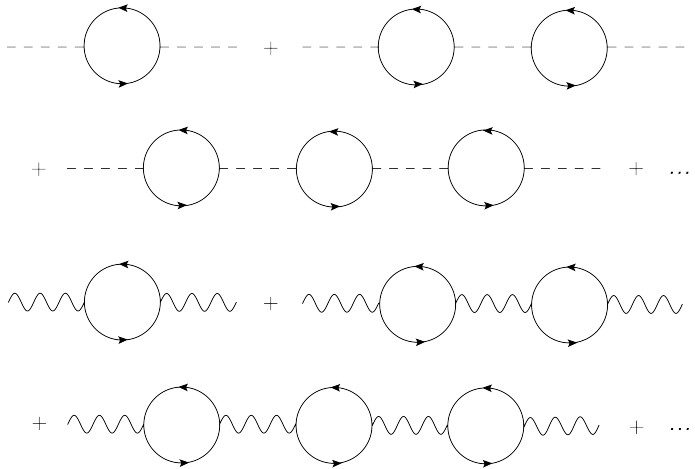

Figure 2: Diagrams that contribute to the boundary propagator of $\partial_y \Phi$ and $A_a$ in the large-$N_f$ limit with $g, h, \lambda, \gamma$ fixed.

The tree-level boundary propagators are (we work in a generic $\xi$ gauge)

$$\langle \partial_y \Phi(p) \partial_y \Phi(-p) \rangle^{(\text{tree})} = -|p| \,,$$
$$\langle A_a(p) A_b(-p) \rangle^{(\text{tree})} = \frac{\lambda/N_f}{1+\gamma^2} \frac{1}{|p|} \left( \delta_{ab} - (1-\xi)\frac{p_a p_b}{|p|^2} + \gamma \epsilon_{abc} \frac{p^c}{|p|} \right). \tag{12}$$

The bubble of $N_f/2$ complex scalars with two scalar insertions is $\frac{g^2}{4|p|}$ and the one with two photon insertions is $-\frac{N_f|p|}{32}\left( \delta^{ab} - \frac{p^a p^b}{p^2} \right)$. The result for the resummed boundary propagators at leading order at large $N_f$ is [17]

$$\langle \partial_y \Phi(p) \partial_y \Phi(-p) \rangle = \frac{-|p|}{1+\frac{g^2}{4}} \,,$$
$$\langle A_a(p) A_b(-p) \rangle = \frac{\lambda/N_f}{\gamma^2 + \left(1+\frac{\lambda}{32}\right)^2} \frac{1}{|p|} \left[ \left(1+\frac{\lambda}{32}\right)\left( \delta_{ab} - \frac{p_a p_b}{|p|^2} \right) + \gamma \epsilon_{abc} \frac{p^c}{|p|} \right] + \frac{\xi \lambda/N_f}{1+\gamma^2} \frac{p_a p_b}{|p|^3} \,. \tag{13}$$

Note that in the limit $g \to \infty$, the field $g\partial_y \Phi$ is identified with the Hubbard-Stratonovich field of the critical $O(N_f)$ model and, consistently, its propagator is finite in this limit and given by $-4|p|$. In the limit $\gamma, \lambda \to \infty$ with fixed $\frac{\gamma}{\lambda} = \frac{\kappa}{2\pi}$ (being $\kappa \equiv k/N_f$), the photon propagator becomes the IR propagator of a 3d Abelian CS gauge field $a$ at level $k$, coupled to $N_f/2$ complex scalars which is

$$\langle a_a(p) a_b(-p) \rangle = \frac{32}{N_f} \frac{1}{1+\left(\frac{16\kappa}{\pi}\right)^2} \frac{1}{|p|} \left( \delta_{ab} + \frac{16\kappa}{\pi} \epsilon_{abc} \frac{p^c}{|p|} \right) + \mathcal{O}(N_f^{-2}), \tag{14}$$

where we have chosen the gauge where there is no $p_a p_b$ term in the tree-level propagator. As expected, if $\kappa = 0$ we recover the result for bosonic $SU(N_f/2)$ QED$_3$ with no CS level, whereas if $\kappa = \infty$ we have the ungauged vector models where the gauge propagator vanishes (the leading term being proportional to $1/\kappa$).

### 2.2.2 Beta functions and anomalous dimensions

The determination of $\beta$ functions and anomalous dimensions at order $1/N_f$ is a standard calculation in large-$N_f$ perturbation theory. We shall recall that bulk quantities do not get renormalized, as UV divergences are local and the bulk is non-interacting.

Following [17] we will adopt a Wilsonian approach and use a hard cutoff $\Lambda$ on the boundary momenta running in loops. In particular, after an RG step in which we integrate out a shell of momenta between $\Lambda$ and $\Lambda' < \Lambda$, the quantities in the UV theory (with cutoff $\Lambda$) and those in the theory with cutoff $\Lambda'$ are related as follows

$$z_{\Lambda'}^m = Z_z^{1/2} z_\Lambda^m \,, \qquad g_{\Lambda'} = Z_z^{-1} Z_g g_\Lambda \,, \qquad h_{\Lambda'} = Z_z^{-3} Z_h h_\Lambda \,. \tag{15}$$

The diagrams that contribute to the wave function renormalization of $z^m$ and to the renormalization of the vertices are depicted in fig. 3.

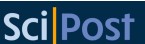

Figure 3: Contribution to the renormalization constants at order $1/N_f$. (a,b) gives the wavefunction renormalization of $z^m$, (c,d) the renormalization of the $g$ vertex, and (e,f) the renormalization of the $h$ vertex (permutations of external legs are omitted).

Writing $Z_{(\cdot)} = 1 + \delta Z_{(\cdot)}$, the result is

$$\delta Z_z = \frac{1}{6\pi^2 N_f} \left( \frac{4g^2}{1 + \frac{g^2}{4}} - \frac{5\lambda(1 + \frac{\lambda}{32})}{\gamma^2 + (1 + \frac{\lambda}{32})^2} \right) \log\left(\frac{\Lambda}{\Lambda'}\right),$$

$$\delta Z_g = \frac{1}{2\pi^2 N_f} \left( -\frac{4g^2}{1 + \frac{g^2}{4}} + \frac{\lambda(1 + \frac{\lambda}{32})}{\gamma^2 + (1 + \frac{\lambda}{32})^2} + \frac{\lambda^2(\gamma^2 - (1 + \frac{\lambda}{32})^2)}{4(\gamma^2 + (1 + \frac{\lambda}{32})^2)^2} \right) \log\left(\frac{\Lambda}{\Lambda'}\right),$$

$$\delta Z_h = \frac{1}{\pi^2 N_f} \left[ \frac{3\lambda^2}{8} \frac{\gamma^2 - \left(1 + \frac{\lambda}{32}\right)^2}{\left(\gamma^2 + \left(1 + \frac{\lambda}{32}\right)^2\right)^2} + \frac{3\lambda}{2} \frac{\lambda\left(1 + \frac{\lambda}{32}\right)}{\gamma^2 + \left(1 + \frac{\lambda}{32}\right)^2} + \frac{18h^2 - 576h - 1536g^2}{64\left(1 + \frac{g^2}{4}\right)^3} \right. \tag{16}$$
$$\left. - \frac{6g^2}{1 + \frac{g^2}{4}} - \frac{g^4}{\left(1 + \frac{g^2}{4}\right)^3 h} \left( \frac{\left(1 + \frac{g^2}{4}\right)\lambda\left(1 + \frac{\lambda}{32}\right)}{\gamma^2 + \left(1 + \frac{\lambda}{32}\right)^2} - \frac{16g^2}{3} \right) \right.$$
$$\left. + \frac{\left(1 + \frac{\lambda}{32}\right)\lambda^3\left(\left(1 + \frac{\lambda}{32}\right)^2 - 3\gamma^2\right)}{6h\left(\gamma^2 + \left(1 + \frac{\lambda}{32}\right)^2\right)^3 \mathbf{8}} \right] \log\left(\frac{\Lambda}{\Lambda'}\right),$$

up to $\mathcal{O}(N_f^{-2})$ corrections. The $\beta$ functions of $g$ and $h$ are given by

$$\beta_g = -\frac{d}{d\log\Lambda}\left(Z_z^{-1}Z_g\, g\right), \qquad \beta_h = -\frac{d}{d\log\Lambda}(Z_z^{-3}Z_h\, h), \tag{17}$$

from which we get, up to $\mathcal{O}(N_f^{-2})$ corrections:

$$\beta_g = \frac{1}{\pi^2 N_f}\left(\frac{8g^2}{3(1+\frac{g^2}{4})} - \frac{4\lambda(1+\frac{\lambda}{32})}{3\left(\gamma^2+(1+\frac{\lambda}{32})^2\right)} + \frac{\lambda^2\left((1+\frac{\lambda}{32})^2-\gamma^2\right)}{8\left(\gamma^2+(1+\frac{\lambda}{32})^2\right)^2}\right)g\,,$$

$$\begin{aligned}
\beta_h = \frac{1}{\pi^2 N_f}\Bigg[&-\frac{9h^3}{32(1+\frac{g^2}{4})^3} + \frac{9h^2}{(1+\frac{g^2}{4})^3} \\
&+\left(\frac{(g^4+8g^2+64)g^2}{2(1+\frac{g^2}{4})^3} - \frac{4\lambda(1+\frac{\lambda}{32})}{\gamma^2+(1+\frac{\lambda}{32})^2} + \frac{3\lambda^2((1+\frac{\lambda}{32})^2-\gamma^2)}{8(\gamma^2+(1+\frac{\lambda}{32})^2)^2}\right)h \\
&-\frac{16g^6}{3(1+\frac{g^2}{4})^3} + \frac{\lambda(1+\frac{\lambda}{32})}{\gamma^2+(1+\frac{\lambda}{32})^2}\left(\frac{g^4}{(1+\frac{g^2}{4})^2} - \frac{\lambda^2((1+\frac{\lambda}{32})^2-3\gamma^2)}{6(\gamma^2+(1+\frac{\lambda}{32})^2)^2}\right)\Bigg]\,.
\end{aligned} \tag{18}$$

From the $\beta$ functions above we can compute the anomalous dimension of $z^\dagger z$ at the leading order in $1/N_f$ and to all orders in the couplings.[5] It is given by (see [17] for a derivation of the relation between the anomalous dimension of singlets and the $\beta$ function of $g$)

$$\gamma_{z^\dagger z} = \frac{\beta_g}{g} = \frac{1}{\pi^2 N_f}\left(\frac{8g^2}{3(1+\frac{g^2}{4})} - \frac{4\lambda(1+\frac{\lambda}{32})}{3\left(\gamma^2+(1+\frac{\lambda}{32})^2\right)} + \frac{\lambda^2\left((1+\frac{\lambda}{32})^2-\gamma^2\right)}{8\left(\gamma^2+(1+\frac{\lambda}{32})^2\right)^2}\right) + \mathcal{O}(N_f^{-2}), \tag{19}$$

which does not depend on $h$ at this order of the large-$N_f$ expansion. Some comments are in order. From eq. (19), in the limit $\gamma,\lambda\to\infty$ with fixed $\frac{\gamma}{\lambda}=\frac{\kappa}{2\pi}$ and $g=0$ (tricritical CS QED$_3$) we get

$$\gamma_{z^\dagger z} = \frac{256(\pi^2-512\kappa^2)}{3N_f(\pi^2+256\kappa^2)^2} + \mathcal{O}(N_f^{-2}). \tag{20}$$

In particular, for $\kappa=0$ we recover the results of [19] for tricritical QED$_3$, while for $\kappa\to\infty$ the anomalous dimension vanishes, as it should since also the gauge field decouples and the theory is free. In the opposite limit, i.e. $\gamma,\lambda\to\infty$ with fixed $\frac{\gamma}{\lambda}=\frac{\kappa}{2\pi}$ and $g=\infty$ (critical CS QED$_3$) we get

$$\gamma_\sigma = -\frac{\beta_g}{g} = -\frac{32(9\pi^4-3584\pi^2\kappa^2+65536\kappa^4)}{3\pi^2 N_f(\pi^2+256\kappa^2)^2} + \mathcal{O}(N_f^{-2}). \tag{21}$$

For $\kappa=0$ we recover the results of ref. [20], while for $\kappa\to\infty$ we recover the results of refs. [21,22] for the critical $O(N_f)$ model.

Concerning the $\beta$ function of $h$, if $g\to\infty$ the dual critical model has a decoupled bulk scalar $\widetilde{\Phi}$ with a Neumann boundary condition. The sextic deformation in (2) corresponds to a cubic deformation $\widetilde{\Phi}^3/\sqrt{N_f}$ on the boundary, with coupling $h'=-h/g^3$. We can easily use this map to find

$$\beta_{h'} = -\frac{18}{\pi^2 N_f}h'^3, \tag{22}$$

---

[5]Note that the boundary condition identifies $\sqrt{N_f}\Phi$ with $gz^\dagger z$ at the boundary. Therefore, except for $g=0$ or $g=\infty$ when the bulk scalar is decoupled, $z^\dagger z$ should not be considered as an independent boundary operator.

for any $\lambda$, as expected. If instead $g = \lambda = 0$, $\beta_h$ reduces to the $\beta$ function of the sextic coupling for $N_f$ real scalars computed in [23]. Finally, in the limit where $g = 0$ and $\gamma, \lambda \to \infty$ with $\kappa$ fixed, it gives the $\beta$ function of the sextic coupling in CS tricritical QED$_3$, which we thoroughly discussed in [18].

## 2.3 Conformal window for bosons

With the $\beta$ functions given in the previous section, the natural step forward is to investigate on the existence of conformal boundary conditions for the theory (2). We therefore look for the real zeros of $\beta_g$ and $\beta_h$, subjected to the condition that the stability bound is satisfied.

The case with no bulk scalar field (so either $g = 0$ or $g = \infty$) was analyzed in [16], where it was shown that under general circumstances such conformal boundary conditions are parametrized by the complex gauge coupling. What happens in the opposite situation, i.e. when the bulk gauge field decouples? If we take $\lambda = 0$ then it is clear from $\beta_g$ in eq. (18) that only for $g = 0$ or $g = \infty$ conformal boundary conditions are possible.[6] These are the familiar Dirichlet and Neumann boundary conditions, which are free.

We are then left to consider $\lambda, \gamma \to \infty$ with $\gamma/\lambda$ constant. By our previous arguments, this limit is bosonic QED$_3$ with $N_f/2$ complex fermions and with CS level $k$, at large $N_f$ and large $k$ with $\kappa = k/N_f$ fixed, coupled to a bulk free scalar with Dirichlet boundary conditions. It is convenient to introduce the variable

$$f_g = \frac{g^2/4}{1 + g^2/4} \in [0, 1], \tag{23}$$

so that in this limit the $\beta$ function for $f_g$ from (18) reads

$$\beta_{f_g} = \frac{64}{3\pi^2 N_f} f_g (1 - f_g) \left( f_g - \frac{8\pi^2(512\kappa^2 - \pi^2)}{(\pi^2 + 256\kappa^2)^2} \right) + \mathcal{O}(N_f^{-2}). \tag{24}$$

Looking for the zeros of $\beta_{f_g}$, beside the ones at $f_g = 0, 1$ where the bulk decouples,[7] we find a family of zeros parametrized by $\kappa$ as

$$f_g = \frac{8\pi^2(512\kappa^2 - \pi^2)}{(\pi^2 + 256\kappa^2)^2}, \tag{25}$$

where $\beta'_{f_g} > 0$ and hence $g$ is marginally irrelevant at these fixed points. A necessary condition for unitary and interacting boundary conditions is that $f_g \in (0, 1)$, which restricts the allowed values of $\kappa$ to be (we take $\kappa \geq 0$ without loss of generality)

$$\frac{\pi}{16\sqrt{2}} < \kappa < \frac{\pi}{16}(\sqrt{5} - \sqrt{2}) \qquad \vee \qquad \kappa > \frac{\pi}{16}(\sqrt{5} + \sqrt{2}). \tag{26}$$

These intervals define unitarity regions which are depicted in orange in fig. 4(a) and 4(b), respectively. At the boundaries of these intervals either $f_g = 0$ or $f_g = 1$.

We also need to analyze the $\beta$ function of $h$ from (18), which in the limit we are considering reads

$$\beta_h = \frac{1}{\pi^2 N_f} \left[ -\frac{9h^3}{32(1 + g^2/4)^3} + \frac{9h^2}{(1 + g^2/4)^3} + \left( \frac{(g^4 + 8g^2 + 64)g^2}{2(1 + g^2/4)^3} + \frac{256\pi^2(\pi^2 - 512\kappa^2)}{(\pi^2 + 256\kappa^2)^2} \right) h \right.$$
$$\left. - \frac{16g^6}{3(1 + g^2/4)^3} + \frac{32\pi^2}{\pi^2 + 256\kappa^2} \left( \frac{g^4}{(1 + g^2/4)^2} - \frac{512\pi^2(\pi^2 - 768\kappa^2)}{3(\pi^2 + 256\kappa^2)^2} \right) \right] + \mathcal{O}(N_f^{-2}). \tag{27}$$

---

[6]Of course if $N_f$ is strictly infinite all $\beta$ functions vanish, and the space of conformal boundary conditions for (2) is spanned by four real parameters: $g, h, \lambda, \gamma$.

[7]If $f_g = 0$ ($f_g = 1$) the theory is given by a bulk photon with Neumann and a bulk scalar with Dirichlet (Neumann) boundary conditions, which are decoupled from CS tricritical (critical) QED$_3$ at the boundary. For a detailed study of the zeros when $f_g = 0, 1$ as a function of $\kappa$, see ref. [18].

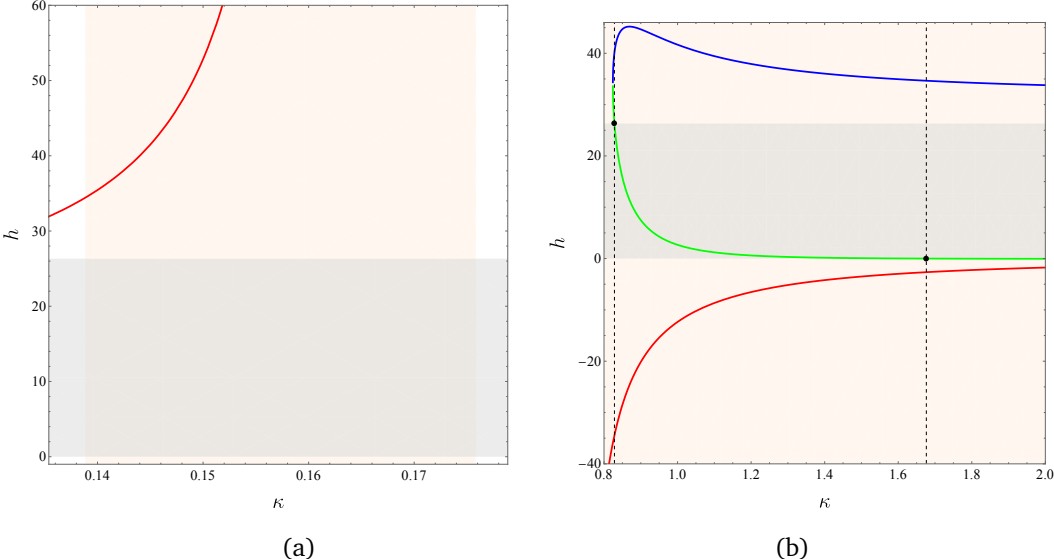

(a)                                          (b)

Figure 4: Interacting, unitary, and stable conformal boundary conditions for the free massless scalar. Different colors correspond to different solutions of $\beta_h = 0$. The unitarity regions of eq. (26), corresponding to real fixed points, are depicted in orange. The fixed points corresponding to stable vacua must lie inside the gray region given by $0 < h < 8\pi^2/3$. The black dots correspond to $(\kappa_-, 8\pi^2/3)$ and $(\kappa_+, 0)$.

Within the region of eq. (26) we find three families of real zeros of $\beta_h$, which are depicted in red, blue, and green in fig. 4. The red and blue families correspond to fixed points of $\beta_h$ where $\beta'_h < 0$ and the sextic operator is marginally relevant, whereas for the green family $\beta'_h > 0$ and hence the sextic operator is marginally irrelevant. Once we combine this with the constraint from vacuum stability (11), depicted in gray in figures 4, we find that unitary and stable interacting conformal boundary conditions for the free massless scalar are possible if

$$\kappa_- \simeq 0.8283 < \kappa < \kappa_+ \simeq 1.6764\,. \tag{28}$$

In this interval, both values of the fixed points of $g^2$ and $h$ are monotonically decreasing function of $\kappa$, ranging from $(g^2, h) \simeq (14.456, 8\pi^2/3)$ at $\kappa = \kappa_-$ to $(g^2, h) \simeq (1.077, 0)$ at $\kappa = \kappa_+$. Since the interacting boundary condition only exists for a finite range of $g^2$, we see that the decoupling limits with critical and tricritical bosonic QED$_3$ on the boundary are not limit points of the family of interacting boundary conditions.

The eigenvalues of the matrix of derivatives of the $\beta$ functions with respect to $g$ and $h$, evaluated at the fixed points, give the anomalous dimensions $\gamma_\pm$ of the classically marginal operators. The result is shown in fig. 5, where we plot $N_f \gamma_\pm$ as a function of $\kappa$ in the conformal window (28). Both anomalous dimensions are strictly positive in this interval, showing that the classically marginal operators are actually marginally irrelevant. Of course there are also strongly relevant singlet boundary primaries in theory, which need to be tuned to zero in order to reach the conformal point. These are: $\Phi$ which has dimension 1, $\partial_y \Phi$ which has dimension 2, and the quartic interaction $(z^\dagger z)^2$ which has dimension 2. Recall that $z^\dagger z$ is not an independent boundary operator, see footnote 5. Let us finally observe that the two curves in fig. 5 intersect at $(\kappa, N_f \gamma) = (0.8350, 0.3799)$, so for the scaling dimensions of the marginal operators we have that, at that point

$$\epsilon \equiv \Delta_+ - \Delta_- = \mathcal{O}(N_f^{-2})\,. \tag{29}$$

As a first sight this result might look suspicious since we might expect the von Neumann-Wigner non-crossing rule [24] to apply to our system, once we regard the dilatation operator

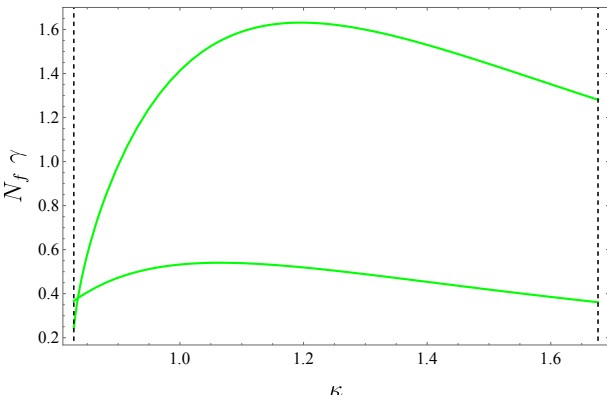

Figure 5: Anomalous dimensions of the classically marginal operators in the conformal window. The dashed black lines denote the boundaries of the interval in eq. (28). The two curves intersect at $(\kappa, N_f \gamma) \simeq (0.8350, 0.3799)$.

as an Hamiltonian on the cylinder [25]. A possible way out is that there is an emergent discrete symmetry at the crossing point. Alternatively, and more likely, the level crossing is an artifact of the large-$N_f$ expansion, signaling that in order to correctly capture the behavior of the spectrum near that point one needs a resummation of the $1/N_f$ expansion. Examples of this phenomenon were considered in [25], in the context of $\mathcal{N} = 4$ SYM with large number of colors and in the extremal spectra of some 3d CFTs. See also [26] for a conformal perturbation theory proof of the non-crossing rule in the context of one-dimensional conformal manifolds. Recent numerical bootstrap investigations of the non-crossing rules can be found in [27] in the context of the Ising CFT in the $4 - \varepsilon$ expansion, and in [28] in the context of $\mathcal{N} = 4$ SYM with finite number of colors.

**More BCFT data in the conformal window**

We conclude this section by computing a few additional observables in the conformal window of section 2.3, and compare with the results from the bootstrap analysis of ref. [9].

The first observable that we shall consider is the anomalous dimension of the leading singlet scalar on the boundary, $\hat{\varepsilon} = z^\dagger z$. It follows from eq. (19) that such anomalous dimension vanishes exactly along the non-trivial solutions of (25). This is of course just a consequence of the "modified Dirichlet" boundary condition, which away from $g = 0$ and $g = \infty$ identifies $\varepsilon$ with $\Phi$ at the boundary. The scaling dimension of $\hat{\varepsilon}$ equals exactly one. This is nicely compatible with fig. 1 of [9].

Next, we compute the anomalous dimension of the leading spin-two primary on the boundary, namely the "pseudo stress-tensor" $\hat{\tau}_{ab}$. We recall that the latter is a symmetric and traceless tensor of spin 2, but it fails to be conserved due to bulk-boundary interactions. In other words its scaling dimension reads

$$\Delta_{\hat{\tau}} = 3 + \gamma_{\hat{\tau}} = 3 + \frac{1}{N_f} \gamma_{\hat{\tau}}^{(1)} + \mathcal{O}(N_f^{-2}), \tag{30}$$

and $\gamma_{\hat{\tau}}^{(1)} \geq 0$ by unitarity. It is not difficult to compute this anomalous dimension using con-

formal multiplet recombination to find[8]

$$\gamma_{\widehat{\tau}}^{(1)} = \frac{\Delta_{g\varphi^2}(3 - \Delta_{g\varphi^2})}{5} \frac{C_{\partial_y \Phi} C_{g\varphi^2}}{C_{\widehat{\tau}}} = \frac{8}{15\pi^2} \frac{g^2}{(1 + \frac{g^2}{4})^2}$$

$$= -\frac{\frac{\pi^2}{512} - \kappa^2}{491520 \left(\kappa^2 + \frac{\pi^2}{256}\right)^4} \left(65536\kappa^4 - 3584\pi^2\kappa^2 + 9\pi^4\right),$$

(31)

where $\kappa$ takes values in the interval (28), $C_\Phi$ and $C_{\partial_y \Phi}$ are the coefficients of the resummed two-point correlation functions of $\Phi$ and $\partial_y \Phi$ in position space, $C_{g\varphi^2} = N_f C_\Phi$, $\Delta_{g\varphi^2} = 1 + \mathcal{O}(N_f^{-1})$ and

$$C_{\widehat{\tau}} = \frac{3N_f}{32\pi^2} + \mathcal{O}(N_f^0),$$

(32)

is the central charge of $N_f$ free scalars. We used that that [17]

$$C_{\partial_y \Phi} = \frac{1}{\pi^2} \frac{1}{1 + \frac{g^2}{4}} + \mathcal{O}(N_f^{-1}) = \frac{65536\kappa^4 - 3584\pi^2\kappa^2 + 9\pi^4}{256^2 \pi^2 \left(\kappa^2 + \frac{\pi^2}{256}\right)^2} + \mathcal{O}(N_f^{-1}),$$

(33)

$$C_\Phi = \frac{1}{2\pi^2} \frac{\frac{g^2}{4}}{1 + \frac{g^2}{4}} + \mathcal{O}(N_f^{-1}) = -\frac{\frac{\pi^2}{512} - \kappa^2}{32 \left(\kappa^2 + \frac{\pi^2}{256}\right)^2} + \mathcal{O}(N_f^{-1}).$$

(34)

Notice that the anomalous dimension (31) is always positive within the conformal window (28), and it vanishes only at the decoupling points for which $g = 0$ and $g = \infty$. Furthermore, it lies well inside the region allowed by the numerical bootstrap bounds for all $N_f \geq 1$, as shown in fig. 6. For the sake of this comparison, we report here the relation between $g$ and the parameter $a_{\Phi^2}$, i.e. the bulk one-point function of $\Phi^2$ with a generic conformal boundary condition

$$a_{\Phi^2} = -\frac{1}{4} + \frac{\frac{g^2}{8}}{1 + \frac{g^2}{4}} + \mathcal{O}(N_f^{-1}) = -\frac{1}{4} + \frac{\pi^2}{32} \frac{\kappa^2 - \frac{\pi^2}{512}}{\left(\kappa^2 + \frac{\pi^2}{256}\right)^2} + \mathcal{O}(N_f^{-1}).$$

(35)

Finally, we shall compute the two-point function of the displacement operator D. We recall that this is a universal boundary primary operator in any BCFT$_d$ that features a bulk stress-energy tensor $T^{\mu\nu}$, as dictated by the (broken) conformal Ward identity [29, 30]

$$\partial_\mu T^{\mu y}(x, 0) = -\delta(y) \mathrm{D}(x),$$

(36)

which fixes the scaling dimension of D to be $d$. For a theory of a massless free scalar $\Phi$, the displacement operator has the following expression

$$\mathrm{D} = \left( \frac{1}{2}(\partial_y \Phi)^2 - \frac{1}{2}(\partial_a \Phi)^2 + \frac{1}{6}\partial_a^2 \Phi^2 \right) \Bigg|_{y=0}.$$

(37)

This computation is rather technical, and we content ourselves with looking at the leading term in the large-$N_f$ expansion (i.e. when $N_f = \infty$), leaving higher-order corrections for future work. The two-point function of D is therefore simply given by Wick's contractions and reads

$$\langle \mathrm{D}(x)\mathrm{D}(0)\rangle = \frac{C_{\mathrm{D}}}{|x|^8} = \frac{\frac{1}{2}\left(C_{\partial_y \Phi}^2 + 4C_\Phi^2\right) + \mathcal{O}(N_f^{-1})}{|x|^8},$$

(38)

---

[8]This can be obtained from the formulae of section 3.2.1 of [9] after observing that we are doing conformal perturbation theory around $N_f$ free scalars in the coupling $1/\sqrt{N_f}$ by the marginal operator $g\,\partial_y \Phi\, \varphi^I \varphi^I$.

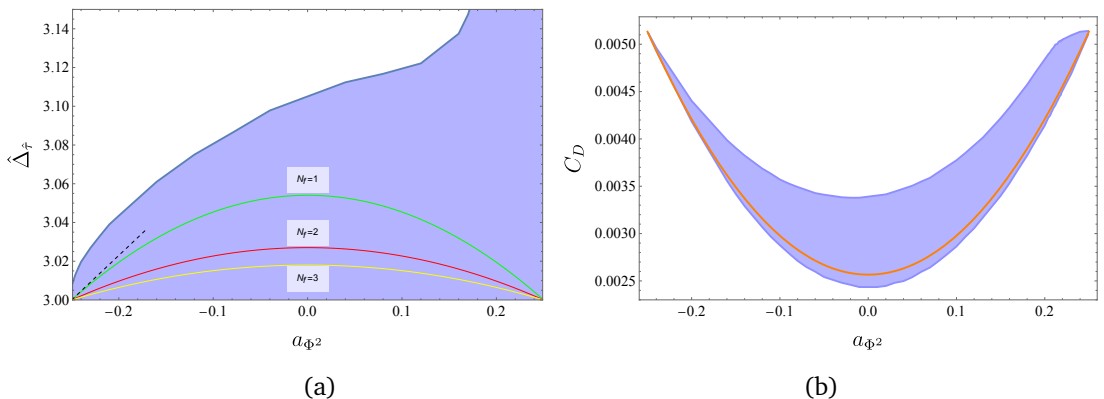

(a)          (b)

Figure 6: Comparison with figures 2 and 4 of [9]. In (a) the different lines correspond to different choices of $N_f$. As in [9], the dashed line is the maximum possible value for $\hat{\Delta}_\tau$ from leading-order conformal perturbation theory, if we assume the Ising model to be the 3d CFT with the lowest central charge.

where $C_\Phi$ and $C_{\partial_y \Phi}$ are the coefficient of the resummed two-point correlation functions of $\Phi$ and $\partial_y \Phi$, in position space.[9] Putting all together we get

$$
\begin{aligned}
C_D &= \frac{1}{2\pi^4} \frac{1 + \frac{g^4}{16}}{(1 + \frac{g^2}{4})^2} + \mathcal{O}(N_f^{-1}) \\
&= \frac{1}{\left(\kappa^2 + \frac{\pi^2}{256}\right)^4} \left( \frac{\kappa^8}{2\pi^4} - \frac{7\kappa^6}{128\pi^2} + \frac{235\kappa^4}{65536} - \frac{127\pi^2\kappa^2}{8388608} + \frac{145\pi^4}{8589934592} \right) + \mathcal{O}(N_f^{-1}).
\end{aligned}
\tag{39}
$$

Notice that $C_D$ interpolates between Dirichlet and Neumann boundary conditions, for which $C_D = \frac{1}{2\pi^4}$ [29,30]. All values in between (in particular those corresponding to the conformal window (28)) happen to lie inside the region allowed by the numerical bootstrap, and in particular close to the lower bound on $C_D$, as shown in fig. 6. As noted in [6], there is a universal prediction for $C_D$ as a function of $a_{\Phi^2}$ coming from the coupling to mean-field theory on the boundary [17, 31, 32], and our calculation at the leading order at large $N_f$ simply reproduces this universal curve as we vary $\kappa$.

## 3 Fermions on the boundary

We consider a 4d bulk scalar field $\Phi$ and bulk Maxwell field $A_\mu$, both with Neumann boundary condition, coupled to $N_f/2$ flavors of 3d Dirac fermions $\chi^m$, where $m = 1, \dots, N_f/2$. The action is

$$
S = S_{\text{bulk}} + \int_{y=0} d^3x \left[ \bar{\chi}^m \slashed{D} \chi^m + \frac{g}{\sqrt{N_f}} \Phi \bar{\chi}^m \chi^m + \frac{h}{\sqrt{N_f}} \Phi^3 \right],
\tag{40}
$$

being $\mathscr{D}_a \equiv \partial_a + iA_a$ the covariant derivative, $\slashed{D} \equiv \gamma^a \mathscr{D}_a$ with $\{\gamma_a, \gamma_b\} = 2\delta_{ab}$ and $S_{\text{bulk}}$ is given in (1). For generic values of the couplings, the continuous part of the boundary global symmetry is $SU(N_f/2) \times U(1)^2$, where again the first factor is a flavor symmetry and the second one comes from the currents $\epsilon_{abc} F^{bc}$ and $F_{ya}$. For $\gamma = 0$ the theory further enjoys parity symmetry, taking $N_f/2$ to be even.

---

[9]We should not worry about the off-diagonal piece, as in the definition of $C_D$ we are holding the operator insertions at separated point.

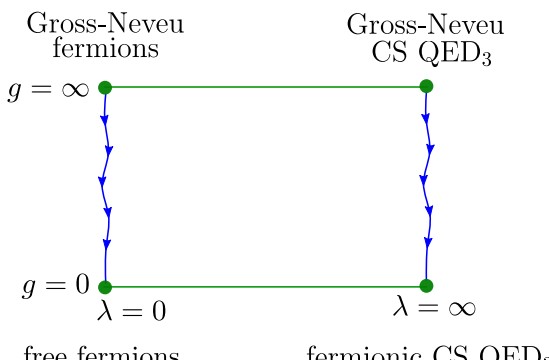

Figure 7: The four 3d fermionic CFTs connected via the interactions to the bulk scalar (vertical lines) and the bulk gauge field (horizontal lines).

As for the bosonic theories discussed earlier, in the large-$N_f$ limit, with $g$, $h$, $\lambda$, $\gamma$ held fixed, the theory (40) interpolates between different bulk/boundary decoupling limits that correspond to different 3d local CFT sectors (see fig. 7):

I. $N_f$ free Majorana fermions for $g = 0$ and $\lambda = 0$.

II. The $O(N_f)$ Gross-Neveu model for $g = \infty$ and $\lambda = 0$, with a cubic interaction $h/g^3$ in the Hubbard-Stratonovich field.

III. Gross-Neveu (GN) QED$_3$ with CS level $k$ and $N_f/2$ flavors of Dirac fermions, for $g = \infty$ and $\lambda = \gamma = \infty$, with $k/N_f = 2\pi\gamma/\lambda$ fixed, and a cubic interaction $h/g^3$ in the Hubbard-Stratonovich field.

IV. Fermionic QED$_3$ with CS level $k$ and $N_f/2$ flavors of Dirac fermions, for $g = 0$ and $\lambda = \gamma = \infty$, with $k/N_f = 2\pi\gamma/\lambda$ fixed.

Here by the Gross-Neveu QED$_3$ we mean the CFT obtained from the $O(N_f)$ Gross-Neveu model by gauging a $U(1)$ subgroup of the $O(N_f)$ symmetry and flowing to the IR, or equivalently a UV fixed point of fermionic QED$_3$ with $N_f/2$ Dirac fermions deformed by an irrelevant quartic interaction.

As in the bosonic case, $\lambda$ and $\gamma$ are always exactly marginal couplings, whereas both $g$ and $h$ develop non-trivial, $\lambda$- and $\gamma$-dependent $\beta$ functions at the subleading order in the large-$N_f$ expansion, and the decoupling limits will be generically connected by RG flows.

## 3.1 Stability of marginal couplings

Next, we discuss the stability bounds on the cubic coupling $h$ in (40). At the classical level we must set $h = 0$ in order for the vacuum to be stable, since the cubic potential would be unbounded from below otherwise. At the quantum level this condition is relaxed, as we now explain. The key ingredient is again the large-$N_f$ effective potential, which at the leading order can be computed by considering the boundary Lagrangian in (40) neglecting the contribution of the gauge field, which is subleading in this limit. We then have

$$\mathscr{L} = \bar{\chi}^m \slashed{\partial} \chi^m + \frac{g}{\sqrt{N_f}} \Phi \bar{\chi}^m \chi^m + \frac{h}{\sqrt{N_f}} \Phi^3 . \tag{41}$$

The only fundamental field that can take a vev is the bulk scalar $\Phi$, hence we let

$$\langle \Phi(x^a, y) \rangle = \sqrt{N_f} U , \tag{42}$$

where $U$ is a vev the scales as $O(N_f^0)$. Upon plugging into (41) and performing the Gaussian integral over the $N_f$ fields we get

$$\mathcal{V}_{\text{eff}}(U) = N_f h U^3 - \frac{N_f}{2} \operatorname{tr} \log \left( \slashed{\partial} + g U \right) + \mathcal{O}(N_f^0). \tag{43}$$

The trace can be computed in dimensional regularization to find

$$\mathcal{V}_{\text{eff}} = N_f \left( h U^3 + \frac{|g|^3}{12\pi} |U|^3 \right) + \mathcal{O}(N_f^0). \tag{44}$$

The gap equation for $U$ is given by

$$3h U^2 + \frac{|g|^3}{4\pi} U |U| = 0. \tag{45}$$

The unique solution to the gap equation, $U = 0$, corresponds a global minimum of the effective potential if

$$|h| < \frac{|g|^3}{12\pi}. \tag{46}$$

In the fermionic case the classical stability region is enlarged by quantum effects. This has to be expected, since fermionic self-interactions are attractive and tend to stabilize the vacuum.

Note that this bound holds for any finite $g$. In particular, for $g = 0$ it is never satisfied, meaning that the cubic boundary interaction has to vanish in order for the theory to be stable. In the limit $g \to \infty$, the bulk scalar field decouples and the resulting 3d theories are the ones where the quartic coupling flows to criticality. Such Gross-Neveu models include a cubic coupling in the Hubbard-Stratonovich field which is given by $y = 2h/g^3$. In that case the bound (46) reads

$$|y| < \frac{1}{6\pi}, \tag{47}$$

in agreement with the stability condition found in [18].

## 3.2 RG analysis

In this section we present the $\beta$ functions for the marginal couplings, as well as the anomalous dimensions for a few operators of the theory in eq. (41), extending the results of [17] to a non-zero $\theta$ term. Feynman rules are collected in appendix A.

### 3.2.1 Exact propagators at large $N_f$

We shall compute the large-$N_f$ boundary propagators of $\Phi$ and $A_a$ by resumming the geometric series of the 1PI bubbles connected by tree-level propagators, as depicted in fig. 2. While the bubble of $N_f/2$ Dirac fermions with two photon insertions has the same expression with respect to the bosonic case, the one with two $\Phi$ insertions is $-g^2|p|/16$. At the leading order the resummed propagators in a generic $\xi$ gauge read

$$\langle \Phi(p)\Phi(-p) \rangle = \frac{1}{1 + \frac{g^2}{16}} \frac{1}{|p|}, \tag{48}$$

$$\langle A_a(p)A_b(-p) \rangle = \frac{\lambda/N_f}{\gamma^2 + \left(1 + \frac{\lambda}{32}\right)^2} \frac{1}{|p|} \left[ \left(1 + \frac{\lambda}{32}\right)\left(\delta_{ab} - \frac{p_a p_b}{|p|^2}\right) + \gamma \epsilon_{abc} \frac{p^c}{|p|} \right] + \frac{\xi\lambda/N_f}{1 + \gamma^2} \frac{p_a p_b}{|p|^3}. \tag{49}$$

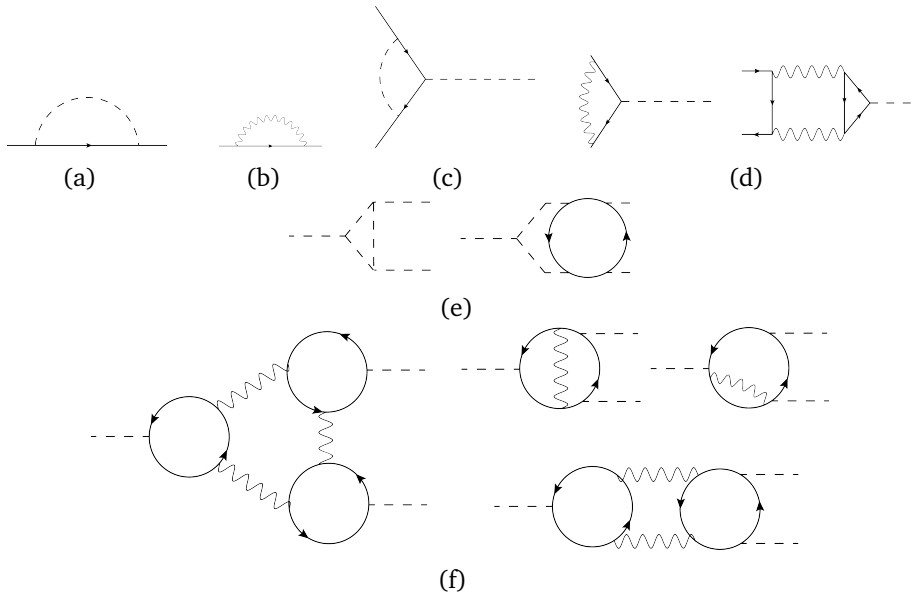

Figure 8: Contribution to the renormalization constants at order $1/N_f$. (a,b) gives the wavefunction renormalization of $\chi^m$, (c,d) the renormalization of the $g$ vertex, and (e,f) the renormalization of the $h$ vertex (permutations of external legs are omitted).

Note that in the limit $g \to \infty$, the field $g\Phi$ is identified with the Hubbard-Stratonovich field of the $O(N_f)$ Gross-Neveu model and, consistently, its propagator is finite in this limit and given by $16/|p|$. In the limit $\gamma, \lambda \to \infty$ with fixed $\frac{\gamma}{\lambda} = \frac{\kappa}{2\pi}$, one gets the IR propagator of a 3d Abelian CS gauge field $a$ at level $k$ coupled to $N_f/2$ Dirac fermions

$$\langle a_a(p)a_b(-p)\rangle_{3d} = \frac{32}{N_f} \frac{1}{1 + \left(\frac{16\kappa}{\pi}\right)^2} \frac{1}{|p|} \left(\delta_{ab} + \frac{16\kappa}{\pi}\epsilon_{abc}\frac{p^c}{|p|}\right) + \mathcal{O}(N_f^{-2}), \tag{50}$$

in the gauge where there is no term proportional to $p_a p_b$. As expected, if $\kappa = 0$ we recover the result for fermionic QED$_3$ with $N_f/2$ flavors and no CS level, whereas if $\kappa = \infty$ we have the ungauged vector models where the gauge propagator vanishes (the leading term being proportional to $1/\kappa$).

### 3.2.2 Beta functions and anomalous dimensions

We will again adopt the Wilsonian approach of section 2 and use a hard cutoff $\Lambda$ on the boundary momenta running in loops. The quantities in the UV theory (with cutoff $\Lambda$) and those in the theory with cutoff $\Lambda' < \Lambda$ are related as follows

$$\chi^m_{\Lambda'} = Z_\chi^{1/2} \chi^m_\Lambda, \qquad g_{\Lambda'} = Z_\chi^{-1} Z_g g_\Lambda, \qquad h_{\Lambda'} = Z_h h_\Lambda. \tag{51}$$

The diagrams that contribute to the wave function renormalization of $\chi^m$ and to At the leading

order in the large-$N_f$ expansion, writing $Z_{(\cdot)} = 1 + \delta Z_{(\cdot)}$, we find (up to $\mathcal{O}(N_f^{-2})$ corrections)

$$\delta Z_\psi = \frac{1}{6\pi^2 N_f} \left( \frac{g^2}{1 + \frac{g^2}{16}} + \frac{\lambda(1 + \frac{\lambda}{32})}{\gamma^2 + (1 + \frac{\lambda}{32})^2} \right) \log\left( \frac{\Lambda}{\Lambda'} \right),$$

$$\delta Z_g = \frac{1}{2\pi^2 N_f} \left( -\frac{g^2}{1 + \frac{g^2}{16}} + \frac{3\lambda(1 + \frac{\lambda}{32})}{\gamma^2 + (1 + \frac{\lambda}{32})^2} + \frac{\lambda^2 \left( \gamma^2 - (1 + \frac{\lambda}{32})^2 \right)}{4 \left( \gamma^2 + (1 + \frac{\lambda}{32})^2 \right)^2} \right) \log\left( \frac{\Lambda}{\Lambda'} \right),$$

$$\delta Z_h = \frac{1}{\pi^2 N_f} \left( \frac{18 h^2}{(1 + \frac{g^2}{16})^3} - \frac{3 g^4}{8(1 + \frac{g^2}{16})^2} + \frac{g^3}{64} \frac{\lambda^2 \gamma (1 + \frac{\lambda}{32})}{h(\gamma^2 + (1 + \frac{\lambda}{32})^2)^2} \right.$$
$$\left. - \frac{g^3}{6 \cdot 8^3} \frac{\lambda^3 \gamma (3(1 + \frac{\lambda}{32})^2 - \gamma^2)}{h(\gamma^2 + (1 + \frac{\lambda}{32})^2)^3} \right) \log\left( \frac{\Lambda}{\Lambda'} \right). \tag{52}$$

The $\beta$ functions of $g$ and $h$ are given by

$$\beta_g = -\frac{d}{d\log\Lambda} \left( Z_\chi^{-1} Z_g \, g \right), \qquad \beta_h = -\frac{d}{d\log\Lambda} Z_h h, \tag{53}$$

from which we get

$$\beta_g = \frac{1}{\pi^2 N_f} \left( \frac{2g^2}{3(1 + \frac{g^2}{16})} - \frac{4\lambda(1 + \frac{\lambda}{32})}{3\left(\gamma^2 + (1 + \frac{\lambda}{32})^2\right)} + \frac{\lambda^2 \left( (1 + \frac{\lambda}{32})^2 - \gamma^2 \right)}{8\left(\gamma^2 + (1 + \frac{\lambda}{32})^2\right)^2} \right) g, \tag{54}$$

$$\beta_h = \frac{1}{\pi^2 N_f} \left( -\frac{18 h^3}{(1 + \frac{g^2}{16})^3} + \frac{3 g^4 h}{8(1 + \frac{g^2}{16})^2} - \frac{g^3}{64} \frac{\lambda^2 \gamma (1 + \frac{\lambda}{32})}{(\gamma^2 + (1 + \frac{\lambda}{32})^2)^2} + \frac{g^3}{6 \cdot 8^3} \frac{\lambda^3 \gamma (3(1 + \frac{\lambda}{32})^2 - \gamma^2)}{(\gamma^2 + (1 + \frac{\lambda}{32})^2)^3} \right),$$

up to $\mathcal{O}(N_f^{-2})$ corrections. From the $\beta$ functions above we can compute the anomalous dimension of $\bar{\chi}\chi$ at the leading order in $1/N_f$ and to all orders in $\kappa$.[10] It is given by

$$\gamma_{\bar{\chi}\chi} = \frac{\beta_g}{g} = \frac{1}{\pi^2 N_f} \left( \frac{2g^2}{3(1 + \frac{g^2}{16})} - \frac{4\lambda(1 + \frac{\lambda}{32})}{3\left(\gamma^2 + (1 + \frac{\lambda}{32})^2\right)} + \frac{\lambda^2 \left( (1 + \frac{\lambda}{32})^2 - \gamma^2 \right)}{8\left(\gamma^2 + (1 + \frac{\lambda}{32})^2\right)^2} \right) + \mathcal{O}(N_f^{-2}), \tag{55}$$

and it does not depend on $h$ at this order of the large-$N_f$ expansion. From this result, in the limit $\gamma, \lambda \to \infty$ with fixed $\frac{\gamma}{\lambda} = \frac{\kappa}{2\pi}$ and $g = 0$ (fermionic CS QED$_3$) we get

$$\gamma_{\bar{\chi}\chi} = \frac{256(\pi^2 - 512\kappa^2)}{3 N_f (\pi^2 + 256\kappa^2)^2} + \mathcal{O}(N_f^{-2}). \tag{56}$$

This result equals the anomalous dimension for $z^\dagger z$ in bosonic tricritical CS QED$_3$, see eq. (20). Note that $\gamma_{\bar{\chi}\chi} = 0$ for $\kappa \to \infty$ as it should, while we recover the result of refs. [33, 34] for $\kappa = 0$. In the opposite limit, i.e. $\gamma, \lambda \to \infty$ with fixed $\frac{\gamma}{\lambda} = \frac{\kappa}{2\pi}$ and $g = \infty$ (Gross-Neveu CS QED$_3$) we get

$$\gamma_\sigma = -\frac{\beta_g}{g} = -\frac{32(9\pi^4 - 3584\pi^2\kappa^2 + 65536\kappa^4)}{3\pi^2 N_f (\pi^2 + 256\kappa^2)^2} + \mathcal{O}(N_f^{-2}), \tag{57}$$

---

[10]Similarly to the discussion in foonote 5 regarding the scalar case, the boundary condition identifies $\partial_y \Phi$ with $\frac{g}{\sqrt{N_f}} \bar{\chi}\chi + \frac{3h}{\sqrt{N_f}} \Phi^2$, and as a result $\bar{\chi}\chi$ is not an independent boundary operator except in the limits $g = 0$ and $g = \infty$ in which the bulk scalar is decoupled. Differently from the scalar case, however, there are now two operators of classical dimension 2 on the right-hand side, so the scaling dimensions at the coupled fixed points involve the possibility of operator mixing. This will be discussed below.

which equals the anomalous dimension of the Hubbard-Stratonovich field in critical bosonic CS QED$_3$, see eq. (21). Note that we recover the anomalous dimension for the Hubbard-Stratonovich field of the $O(N_f)$ Gross-Neveu model [35] for $\kappa = \infty$, while for $\kappa = 0$ we recover the results of refs. [19,36].

Concerning the $\beta$ function of $h$, for $g = 0$ (and any $\lambda$) it reduces to the correct $\beta$ function of the cubic deformation $\Phi^3$ for a bulk scalar with Neumann boundary conditions, see (22). In the limit where $g \to \infty$ the $\beta$ function that stays finite is the one of $y = 2h/g^3$, which corresponds to the cubic coupling in the Hubbard-Stratonovich field for Gross-Neveu models. If we further consider $\lambda = 0$, this matches $\beta_y$ of the ungauged $O(N_f)$ Gross-Neveu model computed in [37]. Instead, if $\gamma, \lambda \to \infty$ with $\kappa$ fixed, it gives the $\beta$ function of the cubic coupling in Gross-Neveu CS QED$_3$, which we thoroughly discussed in [18].

### 3.3 Conformal window for fermions

We are looking for conformal boundary conditions for the massless free scalar $\Phi$ alone. As in the bosonic case, choosing $\lambda = 0$ leaves us with either Neumann or Dirichlet boundary condition for $\Phi$, and no interesting dynamics. We shall instead consider the limit where $\lambda, \gamma \to \infty$ with $\kappa$ constant, corresponding to coupling the bulk free scalar $\Phi$ with Neumann boundary condition to fermionic QED$_3$ with $N_f/2$ Dirac fermions and with CS level $k$, at large $N_f$ and large $k$ (with $\kappa = k/N_f$ fixed). We introduce again the compactified variable

$$f_g = \frac{g^2/16}{1 + g^2/16} \in [0, 1], \tag{58}$$

so that the $\beta$ function for $f_g$ reads

$$\beta_{f_g} = \frac{64}{3\pi^2 N_f} f_g (1 - f_g) \left( f_g - \frac{8\pi^2(512\kappa^2 - \pi^2)}{(\pi^2 + 256\kappa^2)^2} \right) + \mathcal{O}(N_f^{-2}). \tag{59}$$

Note that the $\beta_{f_g}$ above equals the $\beta_{f_g}$ for the bosonic case, see eq. (24). We conclude that, beside the free conformal boundary conditions at $f_g = 0, 1$, where the bulk decouples, we find again a family of zeros of $\beta_{f_g}$ parametrized by $\kappa$ as

$$f_g = \frac{8\pi^2(512\kappa^2 - \pi^2)}{(\pi^2 + 256\kappa^2)^2}, \tag{60}$$

where $\beta'_{f_g} > 0$ and hence $g$ is marginally irrelevant at these fixed points. The unitarity regions corresponding to $f_g \in (0, 1)$, depicted in orange in fig. 9(a) and 9(b), respectively, are again

$$\frac{\pi}{16\sqrt{2}} < \kappa < \frac{\pi}{16}(\sqrt{5} - \sqrt{2}) \qquad \vee \qquad \kappa > \frac{\pi}{16}(\sqrt{5} + \sqrt{2}), \tag{61}$$

where we take $\kappa \geq 0$ without loss of generality. We also need to analyze the $\beta$ function of $h$ from (54), which in the limit we are considering reads

$$\beta_h = \frac{1}{\pi^2 N_f} \left( -\frac{18h^3}{(1 + g^2/16)^3} + \frac{3g^4 h}{8(1 + g^2/16)^2} + \frac{256\pi^3 g^3(3\pi^2 - 1280\kappa^2)\kappa}{3(\pi^2 + 256\kappa^2)^3} \right) + \mathcal{O}(N_f^{-2}). \tag{62}$$

Within the region of eq. (61) we find three families of real zeros for $\beta_h$, which are depicted in red, blue, and green in fig. 9. The red and blue family correspond to fixed point where $\beta'_h < 0$ and the cubic operator is marginally relevant, whereas for the green family $\beta'_h > 0$ and the cubic operator is marginally irrelevant. Combining with the constraint from vacuum stability

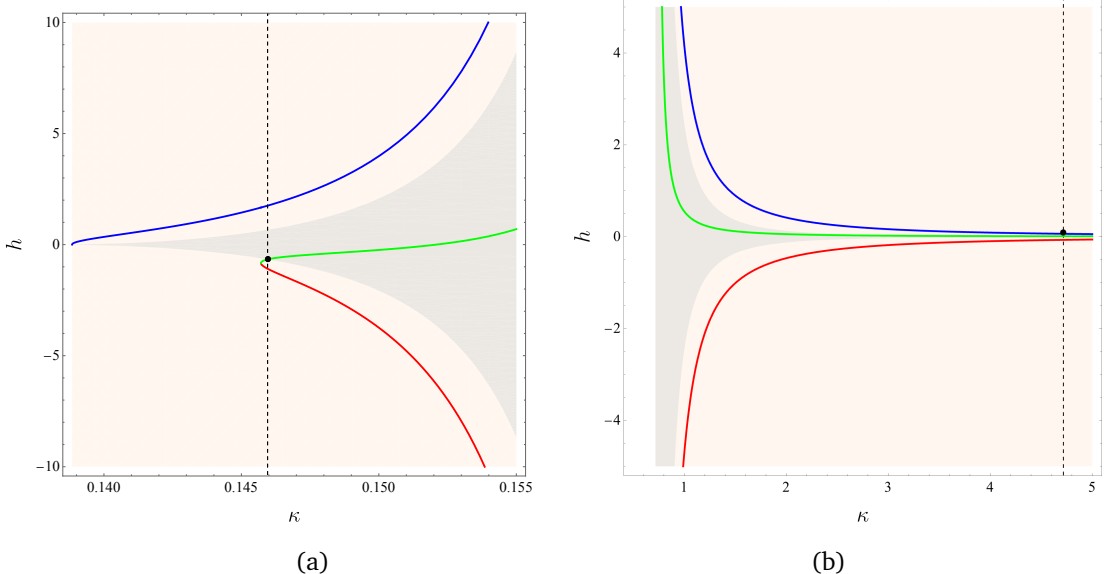

Figure 9: Interacting, unitary, and stable conformal boundary conditions for the free massless scalar. Different colors correspond to different solutions of $\beta_h = 0$. The fixed points corresponding to stable vacua must lie inside the gray region $|h| < g^3/12\pi$, which itself is contained into the unitarity region of eq. (61), corresponding to real fixed points and depicted in orange. In (a), the black dot is at $(\kappa_-, -0.6654)$. In (b), the black dot is at $(\kappa_+, 0.0081)$.

(46),[11] depicted in gray in fig. 9, only one of these solutions corresponds to unitary and stable interacting conformal boundary conditions. They happen when:

$$\kappa_- \simeq 0.1460 < \kappa < \frac{\pi}{16}(\sqrt{5}-\sqrt{2}) \simeq 0.1614 \quad \vee \quad \frac{\pi}{16}(\sqrt{5}+\sqrt{2}) \simeq 0.7167 < \kappa < \kappa_+ \simeq 4.7170\,. \quad (63)$$

In the first interval, both values of the fixed points of $g^2$ and $h$ are monotonically increasing function of $\kappa$, ranging from $(g^2, h) \simeq (8.569, -0.6654)$ at $\kappa = \kappa_-$ to $(g^2, h) = (\infty, \infty)$ at $\kappa = \pi(\sqrt{5}-\sqrt{2})/16$. In the second interval, both values of the fixed points of $g^2$ and $h$ are instead monotonically decreasing function of $\kappa$, ranging from $(g^2, h) = (\infty, \infty)$ at $\kappa = \pi(\sqrt{5}+\sqrt{2})/16$ to $(g^2, h) \simeq (0.4542, 0.0081)$ at $\kappa = \kappa_+$. Hence, the decoupling limit with Gross-Neveu QED$_3$ (but not fermionic QED$_3$) on the boundary is reachable from a limit of the family of interacting boundary conditions, following the green curves of fig. 10.

The eigenvalues of the matrix of derivatives of the $\beta$ functions with respect to $g$ and $h$, evaluated at the fixed points, give the anomalous dimensions $\gamma_\pm$ of the classically marginal operators. The result is shown in fig. 10, where we plot $N_f\gamma_\pm$ as a function of $\kappa$ in the conformal window (63). Both anomalous dimensions are strictly positive in these regions, showing that the the classically marginal operator are actually marginally irrelevant. At either $\kappa = \pi(\sqrt{5}\mp\sqrt{2})/16$ (right endpoint in fig. 10(a) and left endpoint in fig. 10(b), respectively) one operator becomes marginal and $h \to \infty$, but correspondingly the bulk scalar decouples. As for bosons, also here we observe level crossing at a specific value of $\kappa$, given by $\kappa \simeq 1.7806$, where $N_f\gamma_+ = N_f\gamma_- \simeq 0.3310$. Again, there are also strongly relevant singlet boundary primaries in theory, which need to be tuned to zero in order to reach the conformal point. These are: $\Phi$ which has protected dimension 1, $\partial_y\Phi$ which has dimension 2, and a linear combination of $\Phi^2$ and $\bar\chi\chi$, independent from the one which is fixed by the boundary condition, i.e. $\partial_y\Phi = \frac{g}{\sqrt{N_f}}\bar\chi\chi + \frac{3h}{\sqrt{N_f}}\Phi^2$. To determine the linear combination, and its scaling dimension, we

---

[11]We have to plug in (46) the value of $g$ at the fixed point as a function of $\kappa$, computed in (60).

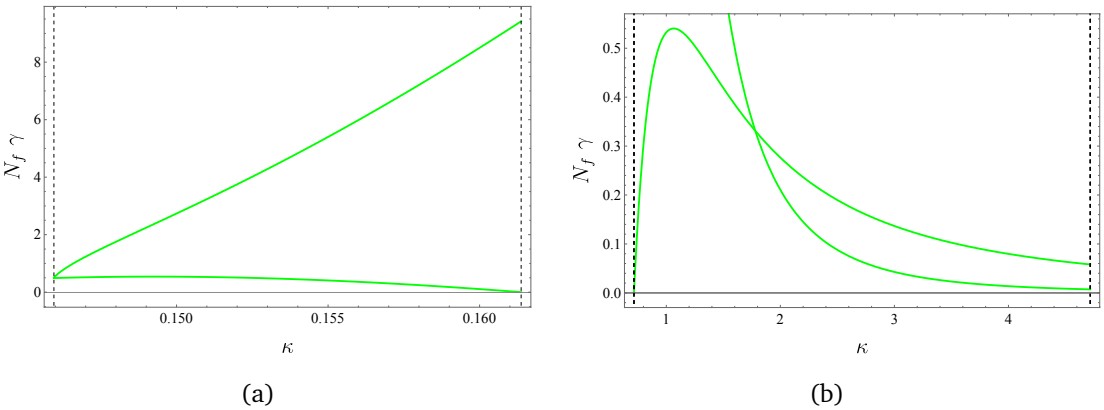

Figure 10: Anomalous dimensions of the classically marginal operators in the conformal window. The dashed black lines denote the boundaries of the intervals in eq. (63). In (a), at $\kappa = \kappa_-$ the two operators are non-degenerate, with small and positive anomalous dimensions, while at $\kappa = \pi(\sqrt{5} - \sqrt{2})/16$ one operator becomes marginal. In (b), one operator becomes marginal at $\kappa = \pi(\sqrt{5} + \sqrt{2})/16$, while at $\kappa = \kappa_+$ the two operators are non-degenerate, with small and positive anomalous dimensions. The two curves intersect at $(\kappa, N_f \gamma) \simeq (1.7806, 0.3310)$.

can adapt the argument used in [17] to determine the anomalous dimension of the operator appearing in the boundary condition. In this case the renormalized operators at scale $\Lambda'$ are defined in terms of those at scale $\Lambda$ as

$$
\begin{aligned}
(\bar{\chi}\chi)_{\Lambda'} &= Z_{11}(\bar{\chi}\chi)_{\Lambda} + Z_{21}(\Phi^2)_{\Lambda}, \\
(\Phi^2)_{\Lambda'} &= Z_{12}(\bar{\chi}\chi)_{\Lambda} + Z_{22}(\Phi^2)_{\Lambda}.
\end{aligned}
\tag{64}
$$

Plugging these equations in the boundary condition

$$
\partial_y \Phi = (Z_{11}g_{\Lambda'} + Z_{12}3h_{\Lambda'})\frac{1}{\sqrt{N_f}}(\bar{\chi}\chi)_{\Lambda} + (Z_{21}g_{\Lambda'} + Z_{22}3h_{\Lambda'})\frac{1}{\sqrt{N_f}}(\Phi^2)_{\Lambda},
\tag{65}
$$

and imposing that $\partial_y \Phi$ does not renormalize, one obtains

$$
\begin{pmatrix} Z_{11} & Z_{12} \\ Z_{21} & Z_{22} \end{pmatrix} \begin{pmatrix} g_{\Lambda'} \\ 3h_{\Lambda'} \end{pmatrix} = \begin{pmatrix} g_{\Lambda} \\ 3h_{\Lambda} \end{pmatrix}.
\tag{66}
$$

Calling $\gamma_{ij} = (\frac{d}{\log \Lambda}Z_{ik})Z_{kj}^{-1}$, we get

$$
\begin{pmatrix} \gamma_{11} & \gamma_{12} \\ \gamma_{21} & \gamma_{22} \end{pmatrix} \begin{pmatrix} g \\ 3h \end{pmatrix} = \begin{pmatrix} \beta_g \\ 3\beta_h \end{pmatrix}.
\tag{67}
$$

These are the two linear relations for the four entries of the anomalous dimension matrix $\gamma_{ij}$, which come from the non-renormalization of $\partial_y \Phi$. Even though they constrain $\gamma_{ij}$, they are not sufficient to determine the two eigenvectors and eigenvalues in terms of the known functions $\beta_g$ and $\beta_h$. However, if we specialize to the case of our interest of the leading non-trivial order $\mathcal{O}(1/N_f)$, the situation simplifies because there are no diagrams contributing to the off-diagonal mixings $Z_{12}$ and $Z_{21}$. As a result up to $\mathcal{O}(1/N_f)$ the matrix $\gamma_{ij}$ is diagonal and we simply get

$$
\begin{aligned}
\gamma_{\bar{\chi}\chi} &= \gamma_{11} = \frac{\beta_g}{g}, \\
\gamma_{\Phi^2} &= \gamma_{22} = \frac{\beta_h}{h}.
\end{aligned}
\tag{68}
$$

In particular, working at this order, both anomalous dimensions vanish at the fixed point, and both $\Phi^2$ and $\bar{\chi}\chi$ have scaling dimension 2. In particular also the linear combination which is not fixed to be $\partial_y\Phi$ has dimension 2.

**More BCFT data in the conformal window**

Finally, we can compute a few observables along the conformal window for fermions: $C_D$, $a_{\Phi^2}^2$, $\gamma_{\hat{\tau}}^{(1)}$. Proceeding as in section 2.3 we find that the leading singlet boundary scalar has scaling dimension $\hat{\Delta}_{\bar{\chi}\chi} = 2$ (as expected from the "modified Neumann" boundary conditions). Furthermore, using that

$$
\begin{aligned}
C_{\partial_y\Phi} &= \frac{1}{\pi^2}\frac{1}{1+\frac{g^2}{16}} + \mathcal{O}(N_f^{-1}) = \frac{65536\kappa^4 - 3584\pi^2\kappa^2 + 9\pi^4}{256^2\pi^2\left(\kappa^2 + \frac{\pi^2}{256}\right)^2} + \mathcal{O}(N_f^{-1}), \\
C_{\Phi} &= \frac{1}{2\pi^2}\frac{\frac{g^2}{16}}{1+\frac{g^2}{16}} + \mathcal{O}(N_f^{-1}) = -\frac{\frac{\pi^2}{512} - \kappa^2}{32\left(\kappa^2 + \frac{\pi^2}{256}\right)^2} + \mathcal{O}(N_f^{-1}), \\
a_{\Phi^2} &= \frac{1}{4} - \frac{\frac{g^2}{32}}{1+\frac{g^2}{16}} + \mathcal{O}(N_f^{-1}) = \frac{1}{4} - \frac{\pi^2}{32}\frac{\kappa^2 - \frac{\pi^2}{512}}{\left(\kappa^2 + \frac{\pi^2}{256}\right)^2} + \mathcal{O}(N_f^{-1}),
\end{aligned}
\tag{69}
$$

as well as

$$
C_{g\bar{\chi}\chi} = N_f C_{\partial_y\Phi}, \qquad \Delta_{g\bar{\chi}\chi} = 2 + \mathcal{O}(N_f^{-1}), \qquad C_{\hat{\tau}} = \frac{3N_f}{32\pi^2} + \mathcal{O}(N_f^0),
\tag{70}
$$

where the latter is the central charge of $N_f/2$ free Dirac fermions, we find

$$
\begin{aligned}
C_D &= \frac{1}{2}\left(C_{\partial_y\Phi}^2 + 4C_{\Phi}^2\right) + \mathcal{O}(N_f^{-1}) = \frac{1}{2\pi^4}\frac{1+\frac{g^4}{256}}{(1+\frac{g^2}{16})^2} + \mathcal{O}(N_f^{-1}) \\
&= \frac{1}{\left(\kappa^2 + \frac{\pi^2}{256}\right)^4}\left(\frac{\kappa^8}{2\pi^4} - \frac{7\kappa^6}{128\pi^2} + \frac{235\kappa^4}{65536} - \frac{127\pi^2\kappa^2}{8388608} + \frac{145\pi^4}{8589934592}\right) + \mathcal{O}(N_f^{-1}),
\end{aligned}
\tag{71}
$$

and

$$
\begin{aligned}
\gamma_{\hat{\tau}}^{(1)} &= \frac{\Delta_{g\bar{\chi}\chi}(3 - \Delta_{g\bar{\chi}\chi})}{5}\frac{C_{\Phi}C_{g\bar{\psi}\psi}}{C_{\hat{\tau}}} = \frac{8}{15\pi^2}\frac{g^2}{(1+\frac{g^2}{4})^2} \\
&= -\frac{\frac{\pi^2}{512} - \kappa^2}{491520\left(\kappa^2 + \frac{\pi^2}{256}\right)^4}\left(65536\kappa^4 - 3584\pi^2\kappa^2 + 9\pi^4\right).
\end{aligned}
\tag{72}
$$

These are the same functions of $\kappa$ that we have found for bosons, however here $\kappa$ is restricted to lie in the interval (63). The anomalous dimensions of the classically marginal operators differ in the two theories.

# 4 Conclusions

In this paper, we have constructed a family of interacting conformal boundary conditions for a four-dimensional free scalar CFT. These conformal boundary conditions are consistent with the currently available bootstrap bounds of [9]. Some of our leading-order predictions, for

example those for $C_{\mathrm{D}}$, while being allowed are also close to saturating the bootstrap bounds. A nice consistency check would be to see that the $1/N_f$ corrections have the correct sign to fit within the bound.

Some interesting common features of the two families of interacting boundary conditions we have constructed are: they both break the $\mathbb{Z}_2$ symmetry that flips the scalar field, they both break the reflection symmetry on the boundary, and they both feature three relevant operators singlet under the boundary global symmetry, i.e. that need to be tuned to reach the BCFT. It is natural to wonder whether any of these is a necessary condition in order for an interacting boundary condition of the free scalar to exist. In particular, even though the free scalar CFT in 4d is expected to describe generic $\mathbb{Z}_2$-breaking second-order transitions, the presence of three relevant boundary deformations makes it hard to find models that realize these boundary transitions, and it would be nice to find examples with less relevant operators. See [12] for condensed matter inspired setups that realize boundary transitions similar to the ones we studied here.

The setup with bulk gauge fields that we studied in this paper provides a way to realize conformal 3d abelian gauge theories, through the decoupling limit, as also emphasized in [16]. An important class of operators that characterizes this theories are the monopole operators, whose scaling dimensions have been studied using various methods, see e.g. [38, 39] and references therein. An interesting direction for the future is to realize the monopole operators in this boundary setup, as boundary endpoints of bulk 't Hooft lines, and compute their scaling dimensions through a new perturbative expansion in the bulk gauge coupling $g$. It remains to be seen whether sensible extrapolations to $g \to \infty$ can be performed and used to estimate these scaling dimensions.

On the more formal side, it is natural to investigate more broadly the space of possible conformal boundary conditions for free CFTs. In the case of the free Maxwell CFT, it is easy to build examples of interacting boundary conditions, and recently they were also studied systematically with the conformal bootstrap in [40]. Some interesting results for the case of free fermions were obtained in [41], and it would be interesting to perform a conformal bootstrap study in that case as well.

# Acknowledgments

We thank C. Behan, I. Salazar Landea, M. Nocchi, B. van Rees and P. van Vliet for discussions and collaborations on related projects.

Views and opinions expressed are those of the author(s) only and do not necessarily reflect those of the European Union or the European Research Council Executive Agency. Neither the European Union nor the granting authority can be held responsible for them.

**Funding information** LD is partially supported by the INFN "Iniziativa Specifica ST&FI". EL is funded by the European Union (ERC, QFT.zip project, Grant Agreement no. 101040260, held by A. Tilloy). PN is supported by the Mani L. Bhaumik Institute for Theoretical Physics and by a DOE Early Career Award under DE-SC0020421.

# A  Feynman rules

## A.1  Dirichlet scalar + Neumann gauge field + $N_f/2$ complex scalars

### A.1.1  Propagators

$$\partial_y \Phi \,\text{------}\, \partial_y \Phi \;=\; -\frac{|p|}{1+\frac{g^2}{4}} \qquad z^m \longrightarrow \bar{z}^n \;=\; \frac{\delta^{mn}}{p^2}$$

$$A_a \,\text{\wavy}\, A_b \;=\; \frac{\lambda/N_f}{\gamma^2 + \left(1+\frac{\lambda}{32}\right)^2} \frac{1}{|p|} \left[\left(1+\frac{\lambda}{32}\right)\delta_{ab} + \gamma\epsilon_{abc}\frac{p^c}{|p|}\right]$$

Figure 11: Large-$N_f$ propagators of $\partial_y \Phi$, $z^m$, and $A_a$ for the Lagrangian (2).

### A.1.2  Vertices

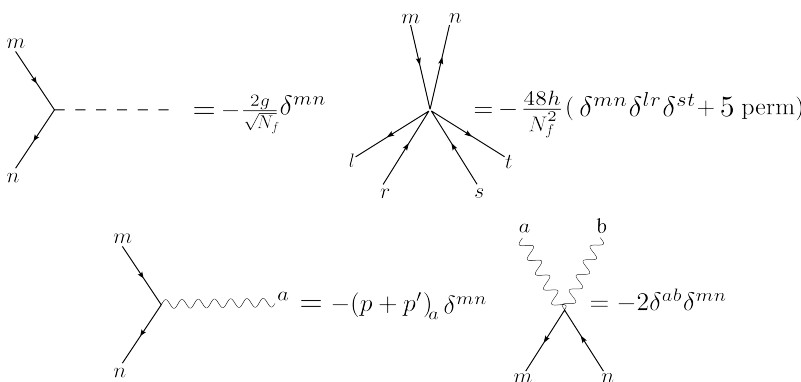

Figure 12: Vertices for the Lagrangian (2). The sextic vertex is completely symmetric in the indices $m, n, l$ and $r, s, t$.

## A.2  Neumann scalar + Neumann gauge field + $N_f/2$ Dirac fermions

### A.2.1  Propagators

$$\Phi \,\text{------}\, \Phi \;=\; \frac{|p|^{-1}}{1+\frac{g^2}{16}} \qquad \chi^m \longrightarrow \bar{\chi}^n \;=\; -\frac{i\slashed{p}\delta^{mn}}{p^2}$$

$$A_a \,\text{\wavy}\, A_b \;=\; \frac{\lambda/N_f}{\gamma^2 + \left(1+\frac{\lambda}{32}\right)^2} \frac{1}{|p|} \left[\left(1+\frac{\lambda}{32}\right)\delta_{ab} + \gamma\epsilon_{abc}\frac{p^c}{|p|}\right]$$

Figure 13: Large-$N_f$ propagators of $\Phi$, $\chi^m$, and $A_a$ for the Lagrangian (40).

### A.2.2 Vertices

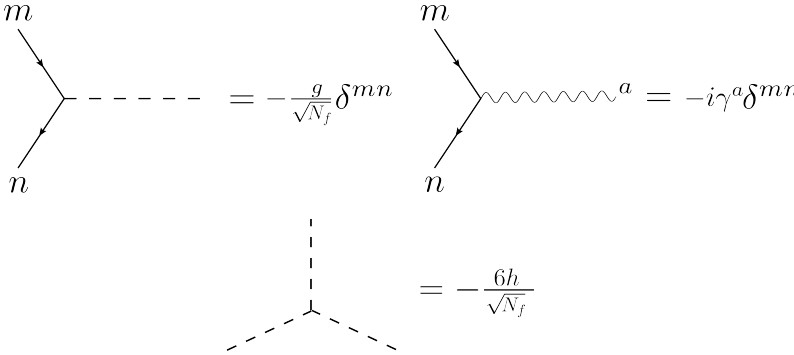

Figure 14: Vertices for the Lagrangian (40).

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
