# Peer review of "Conformal boundary conditions for a 4d scalar field"

_SciPost Physics, doi:SciPost Phys. 16, 090 (2024)_

## Round 1 · Referee Report · Anonymous (Referee 1) · 2024-2-8

Strengths

  1. Clear exposition of useful new results for conformal QED3 on a boundary.

  2. Some nice relations between these perturbative results and previous numerical bootstrap bounds.

Weaknesses

  1. calculation is more incremental progress than drastically new.

Report

This paper analyzes the conformal boundary conditions given by QED3 on the boundary coupled to a 4d bulk. While this theory is strongly coupled, in the large flavor Nf expansion, calculations can be done. The authors analyze the stability of the theory to leading order in 1/N, and also computed various CFT data to leading order in 1/N. They compare some of this data to previous numerical bootstrap bounds, and find that it is in within the bounds, and in some cases close to saturating the bounds.

This paper is a useful contribution to the field of boundary CFT, and certainly deserves to be published.

Requested changes

  1. Can the authors clarify how converged the numerical bootstrap bounds they compare to are, and if in the infinite precision limit they would expect their analytic results to saturate the bounds? In particular, they comment in figure 6 that the quantity C_D is close to saturating the bounds, but if the bounds are not well converged then this could actually be disallowed.

  2. On page 14 the authors discuss how they observe level crossing, which should be forebidden at for a non-perturbative theory, but maybe appear in the large N expansion. They mention previous discussion in the large N limit of SU(N) SYM. Indeed, the large N integrability spectrum does show level crossing, which is resolved at finite N as shown by the numerical bootstrap in recent work.

  3. The calculations in this paper are all at large N, can the authors comment on whether these boundary CFTs are expected to exist at finite N? in particular, in the case of QED3 with no boundary, there is some longstanding controversy in the literature on whether the theories are conformal for small N. Also, is there any reason to expect the large N expansion to be accurate for small N?

  4. Can monopole operators in boundary QED3 be computed at large N? in the non-boundary conformal QED3, these monopole operators are often the lowest dimension operators at small finite N, and so are very important both in experimental realizations, and for comparisons to numerical bootstrap.

  5. are there any ongoing or proposed experimental realizations of boundary QED3, as there are for non-boundary QED3? in the intro, the authors mention how boundary CFTs in general may be relevant phase transitions at the boundary of surfaces, but do not discuss the specific models in question.

  • validity: high
  • significance: good
  • originality: ok
  • clarity: high
  • formatting: excellent
  • grammar: excellent

Author:  Edoardo Lauria  on 2024-02-28  [id 4330]

(in reply to Report 1 on 2024-02-08)

We would like to thank the referee for the time taken to read our paper. We hope the following answer will address referee's question.

1.
In this paper, we are comparing the bootstrap bounds of ref. [9] to a leading-order large-$N_f$ computation for $C_D$. In this limit, the bulk coupling $g$ is exactly marginal, and the theory is described by two coupled GFFs. As discussed in ref. [6], such two coupled GFFs provide an exact solution to the crossing equations of ref. [9]. Therefore, while we do not know if the solution is extremal or not (we have not attempted an extrapolation to infinite derivatives), we know that it can never be excluded by the crossing of ref. [9].

2.
We thank the referee for pointing out a case where it can be shown that level crossing does not occur at the non-perturbative level. In the revised version (now on arxiv) we added a reference to the recent bootstrap analysis of 2312.12576. As we comment below eq. (2.28), we expect level crossing to be an artifact of the large $N_f$ expansion in our case, as well.

3.
Our paper provides evidence for the existence of a family of BCFTs for $N_f$ parametrically large. It is an interesting open problem to understand if such BCFTs persist at finite $N_f$, as well as determining if large $N_f$ results are accurate for small $N_f$.

4.
This is an interesting question. In the revised version (now on arxiv), we added a comment about it in the Conclusions, as well as some references for the monopole operators in conformal QED$_3$.

5.
That is a very interesting possibility. However, as we comment in the Conclusions, the difficulties that one might face experimentally is that the conformal boundary conditions we found require tuning of three relevant boundary deformations. It would be nice to find examples where the number of relevant operators is smaller.

---

## Round 1 · Referee Report · Connor Behan (Referee 2) · 2024-2-20

Report

This paper constructs conformal boundary conditions for a 4d free scalar which have presented more of a challenge than for a 3d free scalar. Section 2 shows that these do not (do) exist when the free boundary conditions are coupled to ungauged (gauged) versions of bosonic Landau-Ginzburg models. Section 3 is entirely analogous and arrives at the same conclusion for fermionic degrees of freedom instead. In both cases, a theta angle is needed for the 3d gauge fields. While this fact was briefly mentioned in a previous paper by the same authors, it is explored in considerable detail here. Another new ingredient is imposing not just unitarity but stability of the vacuum as dictated by the gap equations at large N. When families of fixed points (indexed by Chern-Simons level) satisfying these conditions are found, they are used to compute low-lying conformal data which compare nicely to results obtained with the numerical bootstrap. Although the calculations are very technical, several sanity checks are presented along the way. Another nice aspect is that the various models considered are treated in a unified way by introducing a bulk gauge field and tuning its parameters toward decoupling limits. Since this work makes clear progress on an important problem, I have only minor requests to make before it is published.

Requested changes

  1. The start of section 2 and the start of section 3 both refer to an $SU(N_f/2)$ factor in the symmetry group. Shouldn't this enhance to $SO(N_f)$ since the fields can be split into real and imaginary parts?
  2. Limits II and III in section 3 refer to cubic interactions with couplings given by $h/g^3$. Since these occur for bosons as well, section 2 should probably mention them in the same place.
  3. I would always say "Hubbard-Stratonovich field" instead of "HS field" so people don't accidentally think of "higher spin".
  4. Equations (2.13) and (3.11) use different notation so it is best to pick one.
  5. The $N$ above (3.19) should be $N_f$.
  6. The abstract should say "boundary conditions only exist" with no "s", p15 should say "takes values" with "s" and p27 should pluralize "eigenvector".
  7. One "to" should be removed from "to which the bulk gauge field couples to" on p4 and "saturate" should be "saturating" in the conclusion.

  • validity: high
  • significance: good
  • originality: good
  • clarity: high
  • formatting: excellent
  • grammar: good

Author:  Edoardo Lauria  on 2024-02-28  [id 4329]

(in reply to Report 2 by Connor Behan on 2024-02-20)

We would like to thank the referee for the time taken to read our paper. We hope the following answer will address referee's question.

  1. Due to the interaction with the $U(1)$ gauge field, for generic values of the couplings the flavor part of the global symmetry is really $SU(N_f/2)$. It is indeed enhanced to $SO(N_f)$ in the two 'ungauged' limits: $\lambda=0$, and $\lambda=\infty$ with $k/N_f = \infty$.

  2. The cubic interactions for the Hubbard-Stratonovich fields are generated only in the case of fermionic models with a quartic interaction at criticality, and not for bosonic models. The reason is dimensional analysis: while $\sigma$ has dimension 2 in bosonic models, so that both $\sigma^2$ and $\sigma^3$ are irrelevant, it has dimension 1 in fermionic models, so that $\sigma^2$ is relevant (and triggers a flow from the Gross-Neveu to the free fermion CFTs), while $\sigma^3$ is (classically) marginal. Since we are dealing with parity-breaking theories, no symmetry prevents the coupling $\sigma^3$ to be generated in fermionic theories, and it has to be generically included. In bosonic theories, such term is irrelevant in the critical models, while it has to be included - in the form of a (classically marginal) sextic potential - for tricritical models where the quartic interactions are tuned to zero, as specified in the limits I and IV of section 2.

  3. We agree with the referee and we have modified it accordingly.

4-7. We thank the referee for having spotted the typos. We have fixed them in the revised version (now on arxiv).

---

## Round 1 · Referee Report · Anonymous (Referee 1) · 2024-3-4

Report

The paper is now ready to be published.

---

## Round 1 · Referee Report · Connor Behan (Referee 2) · 2024-3-5

Report

I am happy with the changes and explanation. The paper should now be published.

---

## Editorial Decision

published